# An End-to-End Online Traffic-Risk Incident Prediction in First-Person Dash Camera Videos

Hilmil Pradana 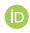

National Institute of Information and Communications Technology, Tokyo 184-8795, Japan; hilmi@nict.go.jp

**Abstract:** Predicting traffic risk incidents in first-person helps to ensure a safety reaction can occur before the incident happens for a wide range of driving scenarios and conditions. One challenge to building advanced driver assistance systems is to create an early warning system for the driver to react safely and accurately while perceiving the diversity of traffic-risk predictions in real-world applications. In this paper, we aim to bridge the gap by investigating two key research questions regarding the driver's current status of driving through online videos and the types of other moving objects that lead to dangerous situations. To address these problems, we proposed an end-to-end two-stage architecture: in the first stage, unsupervised learning is applied to collect all suspicious events on actual driving; in the second stage, supervised learning is used to classify all suspicious event results from the first stage to a common event type. To enrich the classification type, the metadata from the result of the first stage is sent to the second stage to handle the data limitation while training our classification model. Through the online situation, our method runs 9.60 fps on average with 1.44 fps on standard deviation. Our quantitative evaluation shows that our method reaches 81.87% and 73.43% for the average F1-score on labeled data of CST-S3D and real driving datasets, respectively. Furthermore, the proposed method has the potential to assist distribution companies in evaluating the driving performance of their driver by automatically monitoring near-miss events and analyzing driving patterns for training programs to reduce future accidents.

**Keywords:** advanced driver assistance systems; traffic-risk prediction; online videos; unsupervised learning; supervised learning

## 1. Introduction

The growth of Advanced Driver Assistance Systems (ADASs) has the potential impact to improve the global industries engaged in transportation by generating cheaper, faster, and safer architecture [1–5]. The ADAS has been a significant automotive industry focus in recent years. Many researchers developed different problems regarding ADASs, such as effective learning of driver fatigue [6], enhancing traffic sign recognition [7,8], and driving behavior [9]. However, ensuring the driver's safety is a significant challenge, as the assistance must handle various driving scenarios and conditions. ADAS also requires extensive testing and validation to ensure the vehicles can operate safely in different scenarios. In other words, a key challenge to building ADASs is to safely react and accurately perceive the diversity of traffic-risk prediction in real-world applications. Nowadays, the traffic-risk incident is attractive research due to the limitation of data resources and distributions [5,10,11]. For guaranteeing a safe driving strategy, the driving scenario obeys all incident cases with a long-tailed distribution, such that a minimal number of common situations makes up the vast majority of what a driver encounters and a virtually infinite number of rare scenarios [2].

When discussing traffic risk, it is essential to acknowledge that any traffic incident can potentially cause harm [12]. Whether it is a minor fender bender or a severe collision, the outcome can range from minor injuries to life-threatening situations. Furthermore, accidents result in actual injury or damage, representing instances where undesirable

consequences have already materialized, leading to varying degrees of harm to the people, property, or environment [13]. It is an unfortunate event when the potential dangers inherent in an incident have materialized, leading to actual harm. Accidents testify to the critical importance of proactive safety measures, as their occurrence highlights the risks and vulnerabilities within a system. On the other hand, near misses have the potential to cause harm or damage, narrowly avoiding the catastrophic outcome associated with accidents [14]. Near misses can be seen as fortunate occurrences where a combination of factors, including human intervention, luck, or unforeseen circumstances, prevents the situation from escalating into an accident.

Traffic-risk incidents create dangerous situations that may lead to a collision between the ego-vehicle and other moving objects without any notice [15]. We suppose an incident occurs unexpectedly, such as a sudden collision or a pedestrian entering the road. In that case, it may be more difficult for the driver to respond promptly and effectively. Analyzing such incidents is a crucial step toward avoiding dangerous situations in ADASs. In practical application, an incident prediction model aims to tell the driving system if an incident will happen if there is no action from either the ego-driver or other moving objects, such as pedestrians, cyclists, or vehicles. Building an incident prediction model is a more challenging problem due to having to predict what will happen in current driving situations. There are several ways to predict what will happen, such as generating a future trajectory for each object [16], using graph relation learning [17], and deep reinforcement learning [18]. In this paper, we aim to bridge the gap by investigating two key research questions: is the current status safe for the driver to drive in online videos? What types of other moving objects lead the ego-vehicle into a dangerous situation? These questions will lead to a visually explainable model associating the drivers' visual attention and the current situation on the road to detect and predict the current driving status and analyze other moving objects. Unlike other works, we apply the visual attention behavior from successful object saliency to identify and highlight the most noticeable or relevant objects.

Based on recent works [16–22], detecting and predicting traffic incidents in the first person is far from being solved due to the following challenges. First, the visual view of a dashboard-mounted camera's video gives some clues to train a discrimination model before accidents happen. In practice, the accident is too difficult to be captured because of noisy video data and the limited view angle of the dashboard-mounted camera. Previous works use object detection to learn graph relational learning [17] or explicitly use the visual attention behavior to look at incidentally risky regions [18]. In this paper, we propose combining visual attention behavior and object tracking to learn the current driver's status on the road using unsupervised learning to address where precisely the accident's risky regions will happen.

Secondly, event classification is a challenging problem due to video quality, environmental conditions, and the distance between the ego-vehicle and other moving objects. Most researchers use supervised learning [23–25] to classify the events in a public dataset. Other works applied weakly supervised methods [26,27] to detect the abnormality in the video. In this paper, we follow two stages: in the first stage, unsupervised learning is applied to collect all suspicious events before incidents happen from dash camera video of the actual driving. We then use supervised learning to classify the common events before incidents happen from the first stage, where exploration and exploitation can dynamically balance a driving environment. The successful research result from supervised learning [24] can give the advantage of recognizing the class type in the videos. This method transfers info to the general problem to classify the upcoming traffic-risk incidents and generates the general model to be tested in real traffic-risk incident collections. Classification of the common event before near-miss or incidents occur is a crucial problem where we must train our model with the risk annotation before the incident occurs. To solve this problem, we used the contribution of [14], which proposed a new definition for annotating the incident class and annotated the state-of-the-art traffic risk incident dataset [28] using the definition focused on in first-person to recognize upcoming incidents and explore future possibilities.

The proposed approach differs from several existing works [24,29,30] formulated within the unsupervised and supervised learning frameworks. Our proposed system synchronizes all features from dash camera video to extract object tracking [29] and visual attention [30]. We then applied supervised learning to classify the common event type to enrich the classification type. By successfully using supervised learning, our architecture used the model proposed by [24] to classify the traffic risk incident results list from the first stage. The minimum safety distance can be obtained by correlating the current speed estimation from the real-time geographic data and the distance between the ego-vehicle and other moving objects. In the next stage, we classify the suspicious event before near-miss or incidents happen. Then, it is mixed with metadata from the first-stage result to enrich the event classification types. To define the current status of drivers, the safety distance between the ego-vehicle and other moving objects is used to provide a clue about what kind of situation it is.

From the following explanation, we summarize the main contributions of this paper as follows:

- We proposed two-stage algorithms based on unsupervised and supervised learning to predict traffic-risk incidents from dash camera videos based on the human perception of driving.
- We extracted the information from dash camera videos based on a combination of visual driver attention and object tracking of other moving objects appearing on the video to localize where precisely the accident's risky regions will happen and to calculate a safety distance between them for predicting traffic incidents from a dash camera.
- We classified unsafe event collections produced by the first stage using supervised learning and mixed them with the metadata to enrich the classification event type by bringing the metadata from the first stage result for long-term capturing driving events to handle limited data annotation.

## 2. Related Works

This section will discuss previous incident detection and prediction research and the dataset used. The previous research provides a foundation for our research and has significantly contributed to our understanding of incident detection and prediction. Research on incident detection and prediction is a crucial issue in improving road safety and reducing the number of traffic incidents. To deeply understand the current issues in previous research on developing new technologies and algorithms, we summarize the recent research and their weaknesses to give insight for future research.

### 2.1. Incident Dataset

In incident detection and prediction, most previous studies have relied on two types of datasets, closed-circuit television (CCTV) cameras, and dash camera videos. CCTV camera video is often used for traffic monitoring and surveillance and can be found in various settings such as intersections, highways, and bridges. It is used to identify individuals involved in the incident, provide evidence for legal proceedings, assist in investigations, monitor traffic flow, and identify congestion points. It can be used in real time to alert traffic control centers to incidents and adjust traffic signals accordingly. Based on the CCTV camera dataset, the TAND dataset [22] contains many diverse near incidents and perspectives of intersection surveillance videos. The dataset includes three types of video data: drone footage of intersections from a top-down view, real traffic videos recorded using omnidirectional fisheye cameras, and video data simulated using a game engine.

On the other hand, dash camera video can provide visual evidence of an incident on the road and capture the footage from the driver's perspective and the view of the road ahead. These videos are increasingly popular among researchers to identify individuals involved in the incident, provide evidence for legal proceedings, and assist in investigations. Recently, creating a dataset from dash camera video [19,21] has been active research to answer the

research question of what and where incidents will happen from the driver's perspective. Datasets from dash camera videos, such as Driver Anomaly Detection (DAD) [19] and AnAn Accident Detection (A3D) [21], are available publicly. The DAD dataset contains the normal driving class and several unseen anomalous driving classes from data training. The videos in the DAD dataset are recorded with a high frame rate, providing high temporal resolution and allowing for detailed analysis of driver behavior with different multi-modal views. The A3D dataset contains accidents and other on-road incidents and is recorded from the perspective of dashboard cameras in different vehicles. The dataset includes 1500 video clips, each containing an abnormal event at a different point in time. Three human annotators have labeled the start and end times of the anomalies in each video. They were instructed to use common sense to determine when the accident was inevitable and when all participants recovered. Both dash cameras and CCTV videos help to study incident detection and prediction because they provide a large amount of data that can be used to train and evaluate deep learning models. In conclusion, dash cameras and CCTV datasets have been critical in advancing incident detection and prediction research.

*2.2. Incident Detection and Predictions*

Incident detection and prediction provide early warnings and mitigate the impact of incidents on people or property. One of the methods to predict the incident is AdaLEA [20], which enables the model to learn to anticipate accidents earlier as training progresses by assigning penalty weights based on how early the model can predict the accident. However, its method did not report their performance by mixing the testing data with the normal video or real driving scenario with rare incidents. Another method, Future Object Localization (FOL) [21], used an unsupervised deep learning framework for traffic accident detection from egocentric videos. Its method detects significant deviations between the predicted and actual ego-motion as trajectory detection for classifying whether the ego-vehicle is involved in the accident or is just an observer. By constructing temporal relationships using the recurrent neural network by considering spatial relationships and agent-specific features, Ustring [17] integrates Bayesian deep neural networks into the model to handle predictive uncertainty. The Bayesian formulation leads to a ranking loss based on epistemic uncertainty, improving the quality of the learned relationship features and improving performance. Considering the overall guidance of all hidden states during training, a self-attention aggregation layer provides a video-level loss enhancing the model's performance. The Deep Reinforcement Learning (DRL)-based solution proposed by [18] is superior to supervised learning in that DRL can use current observations to achieve a long-term objective, such as making early decisions to prevent future accidents. By detecting vehicle accidents or abnormalities based on a novel frame-level accident detection method using spatiotemporal feature encoding with a multilayer neural network, Ref. [31] follows a coarse-to-fine detection process. In this process, the two temporal features of the frames, a histogram of optical flow and temporal ordinal features, are first encoded as an earthly coding matrix using a multilayer neural network, which is then used to cluster the frames. From the resulting frame clusters, the border frames are detected as potential accident frames. Then, the convolutional neural network features and spatial relationships of the objects detected in these potential accident frames are used to confirm whether they are indeed accident frames. Conditional Style Translation-Separable 3-Dimensional Convolutional Neural Network (CST-S3D) is proposed by [14], where augmented data from the original video is applied to enhance the performance of the classification method.

Using the CCTV dataset, Ref. [22] proposed a vision-based two-stream system for detecting near accidents in real-time using real-time object detection and multiple object tracking. The input video is decomposed into spatial and temporal components. The spatial stream encompasses the appearance information of scenes and objects in the video, while the temporal stream holds information on the motion of moving objects. A multimedia sensing application with alert light and sound systems to detect automobile accidents is reported in [32]. This method sends alerts to other vehicles using roadside sensors without

requiring any changes to the vehicles themselves. Research on accident detection using dashcam video is a more challenging problem based on state-of-the-art results due to relative motions between the ego-car and other vehicles. Unlike CCTV video, which has absolute motion patterns, relative motion patterns have the relationship between a pattern in the camera view causing camera position and driver movement.

## 3. Methods

The proposed method applies a tracking algorithm with the safety distance estimation rather than predicting the future bounding box for other moving objects since, unlike previous work [21], we focus on predicting only the ego-involved and are not concerned with incidents uninvolving the driver. In our assumption, the safety distance follows the human perception during driving. When driving faster, the attention of the distance to other moving objects will be longer so that it is correlated to two different objects since we cannot decide what will happen if only predicting the future bounding box without focusing on the current situation of the driver itself. The detailed explanation and formulation of the proposed framework on traffic-risk incident prediction in first-person dash camera videos are described in Figure 1, where extraction and combination features such as human visual thinking while driving will improve the performance of the proposed method to detect accidents in different conditions. Our framework adopted a heuristic algorithm and supervised learning to exploit the performance to classify the video before traffic incidents happen. On the other hand, the output of the first stage is transferred to the second stage, which is responsible for classifying the event as one of the predefined event types. These events can be classified into common event types with a combination of the metadata from the first stage to enrich the classification types.

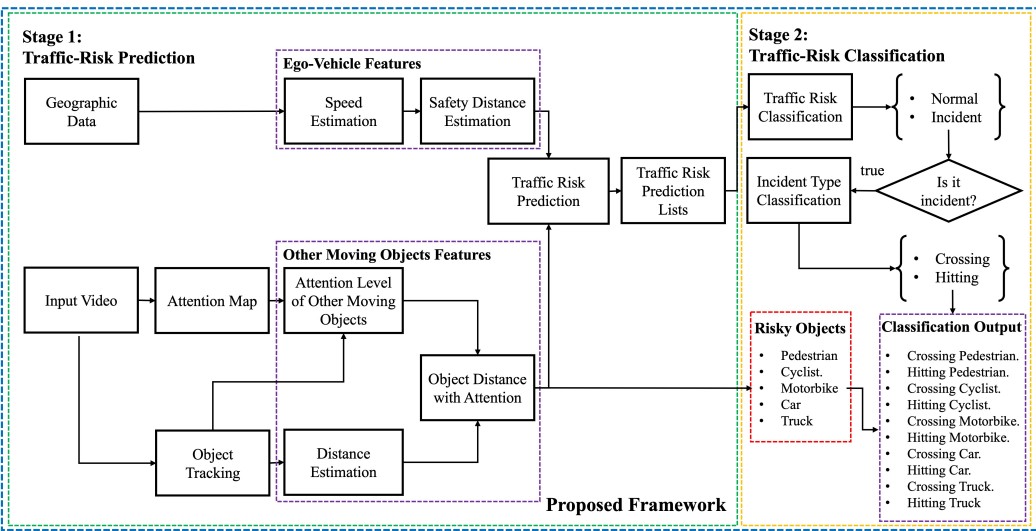

**Figure 1.** The proposed method architecture is depicted in this diagram. In this architecture, we have followed two-stage algorithms for predicting the risky event caused by five types of moving objects together with common event types.

### 3.1. Proposed Method Overview

Our formulation is based on a heuristic technique where extracting some standard features from existing methods [24,30,33] is needed to discover ego-vehicle and other moving object features and adjust the alarm from the traffic risk prediction model. To identify other moving objects and understand the surrounding situations, a tracking bounding box for each object, provided by [33], is required for further processes, where the $x$ and $y$ position of bounding box with $w$ width and $h$ height for the object id $v$ are defined as $B = (b_v, b_x, b_y, b_w, b_h)$. This tracking algorithm can be used to record and analyze traffic pattern data, which can improve traffic flow and safety.

### 3.2. Traffic Risk Prediction

Traffic risk prediction is a rapidly growing field using data and analytical models to estimate the likelihood of various traffic incidents, such as collisions, delays, or congestion. The advantage of traffic risk prediction using unsupervised learning is that there is data limitation to train both the common incident types and risky objects. The primary data source used in our method uses the dash camera video with geographic data. From this data, we can identify patterns in traffic behavior, which can then be used to predict future traffic risks. At the end, the output of this process is traffic risk prediction lists and the metadata of risky objects, which can give useful information or clues to understand how the events will happen. Given $V(1, n) = \{f_1, f_2, f_3, \ldots, f_n\}$ as a raw data source from dash camera videos with $n$ frames, to find and extract the video before the incident happened into $s$ snippets is denoted by:

$$\varsigma(\beta, s) = \{V(\beta_1, \beta_1 + \tau_1), V(\beta_2, \beta_2 + \tau_2), \ldots, V(\beta_s, \beta_s + \tau_s)\} \tag{1}$$

where $\beta_s \in [1, n - \tau_s]$ is traffic risk event time on the $s$-th snippet, and $\tau$ represents the length of the $s$-th snippet from traffic risk video $V$. To get the case of traffic list events, we create two rules to imitate how human drives by following the two standard rules:

1. The driver's perspective focuses more on specific areas than others.
2. If one of the moving objects is closer to the driver than their attention, it must be a near miss or incident if no action is taken.

By following these rules, extracting a snippet of traffic risk $V(\beta_s, \beta_s + \tau_s)$ is denoted by:

$$
\begin{aligned}
D_e^{\beta} &\geq D_{o_k}^{\beta^r}, \\
o_k &\in \zeta_{B^k}^{\beta},
\end{aligned}
\tag{2}
$$

where $D_e^{\beta}$ and $D_{o_k}^{\beta}$ are the safety distance of the driver and the distance of $k$ moving objects to the driver, respectively. $k$ objects belong to the attention map $\zeta_{B^k}^{\beta}$ on frame $f_{\beta}$. By determining the standard rules of how to drive, the extraction features from the driver and other moving objects are automatically used to solve the current problems.

3.2.1. Ego-Vehicle Features

Given geographic data $G_m$ with $m$ sequence point locations, normalizing the length to frame $n$ is the following:

$$\chi = \{r : r = 1 + \lfloor \frac{m}{n} \rfloor \times \eta, \ \eta \in n\}, \tag{3}$$

Then, to compute speed estimation $\varphi_n$ on the $n$-th frame from the geographic data, $G_r$ is calculated by:

$$\varphi_n = \frac{\varrho(G_r, G_{r+1})}{\psi}, \tag{4}$$

where $\varrho$ is the calculation method provided by [34], $\psi$ is represented in frame per second of the video, and $\varphi_n$ is the speed estimation in $m/s$.

Besides that, safety distance estimation $D_e^n$ is represented by the minimum distance the driver can reach when other objects have suspicious movements, such as suddenly stopping. The relationship between $\varphi_n$ and $D_e^n$ can be explained by the following:

$$D_e^n = \frac{\varphi_n{}^2}{2\mu g}, \tag{5}$$

where $g$ is acceleration due to gravity with constant value 9.81 m/s$^2$. $\mu$ is defined by the mean coefficient of friction [35] that can be adjusted as 0.8. From their relation, both $\varphi_n$ and $D_e^n$ are the braking distance at a reasonable distance based on an estimate of the visibility [36]. Several factors related to the vehicle, road, and driver behavior influence a vehicle's braking distance. Distance and time of braking are inversely proportional, and the process behavior changes at various distances, braking times and given powers.

### 3.2.2. Other Moving Object Features

Two features were extracted from other moving objects: estimation of the distance from other objects to the driver and the driver's attention level on the videos. To estimate the distance $D_{o_k}^{\beta^r}$, we calculate the comparison between the actual size and appearing size of each object by following:

$$D_{o_k}^{\beta^r} = D_{o_k}^{\beta^p} * \frac{S_{o_k}^{\beta^r}}{S_{o_k}^{\beta^p}},$$

(6)

where $S_{o_k}^{\beta^r}$ and $S_{o_k}^{\beta^p}$ are actual and appearing sizes of other moving objects in videos. $D_{o_k}^{\beta^p}$ is the distance from these objects to the ego-vehicle in the frame. Then, paying attention to the potential incidents, such as vehicles entering the driver's lane or pedestrians crossing the street, it is a crucial problem to localize what kind of other moving objects can lead the driver to collision by using the driver's attention level. The driver's attention can be determined through visual saliency prediction.

To estimate attention level $\xi_{B^k}$ from the highest attention level $\xi_{B^1}$ to the lowest attention level $\xi_{B^k}$, we are given the bounding box $B^k$ of $k$ moving objects, which is calculated by the following:

$$\xi_{B^k}^{\beta} = \frac{\sum_{q=b_x^k}^{b_x^k+b_w^k} \sum_{r=b_y^k}^{b_y^k+b_h^k} Att(f_\beta(q,r))}{b_w^k \times b_h^k},$$

(7)

where $Att(f_\beta)$ is the attention level in a two-dimensional image of frame $f_\beta$ by computing from the model provided by [30].

After obtaining the absolute distance between the driver and other moving objects $D_{o_k}^{\beta^r}$ and $\xi_{B^k}^{\beta}$, and attention levels for $k$ moving objects, we adjust how the attention can be changed simultaneously. Based on the behavior of those perspectives, we have the changing attention rule:

1.  If the attention map indicates the moving object constantly moving away from the ego-vehicle for a continuous duration of three seconds, there will be a shift in the focus of attention towards another object. Throughout this period, the driver and moving object will try to avoid a potential collision, as illustrated in Figure 2, implying that the object's movement and trajectory will be monitored. If it is observed that it is moving away for an extended duration, then it will no longer be considered a priority for attention. Instead, attention will be redirected toward objects requiring more immediate attention, such as those closer to the ego-vehicle.
2.  The focus of attention will be shifted when the object being monitored by the attention map is no longer being tracked. This can happen when the object disappears from the dash camera video view, as shown in Figure 3. In such cases, the attention map will stop displaying information about the object and instead shift its focus to other moving objects currently within the camera's view.

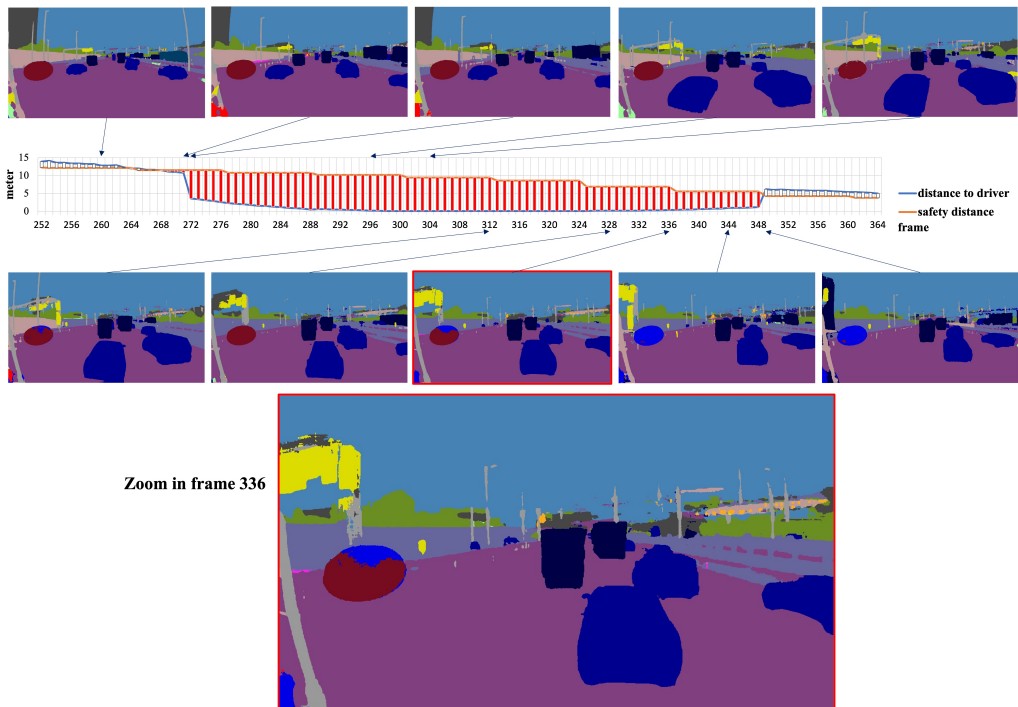

**Figure 2.** Illustration of the clipped temporal window during the near-miss with another moving object. In this situation, the safety distance is lower than the distance of its object to the driver shown on the red bar, where this situation is risky with a car.

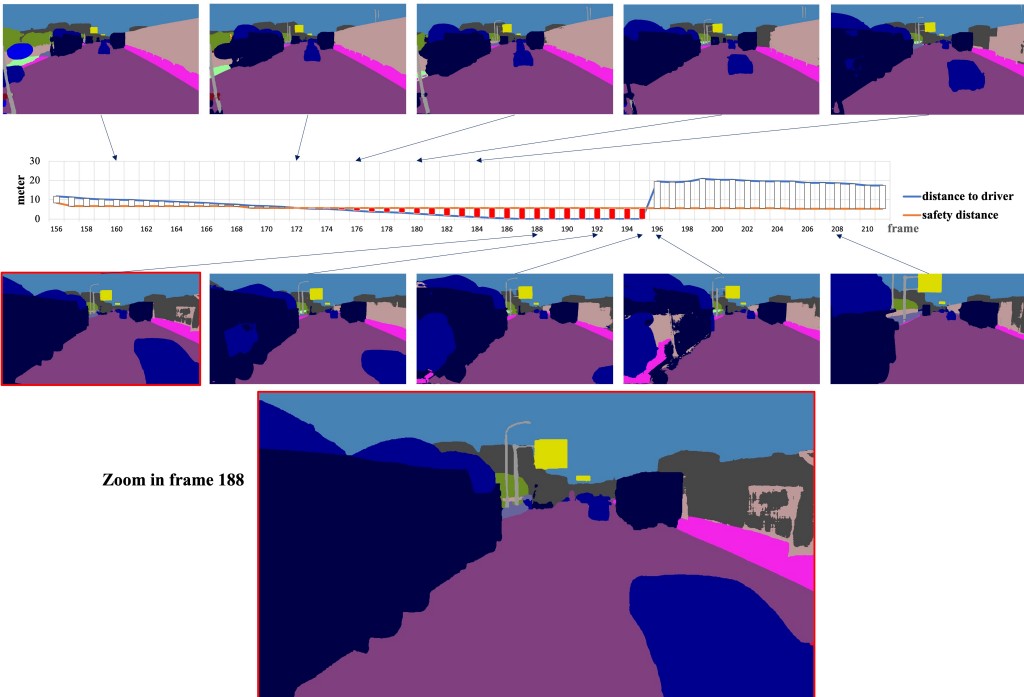

**Figure 3.** Illustration of the clipped temporal window during the near-miss with another moving object. In this situation, the safety distance is lower than the distance of its object to the driver shown on the red bar, where this situation is risky with a car.

### 3.3. Traffic-Risk and Incident Type Classification

To classify the output generated by the decision process of traffic risk prediction, the proposed method utilizes supervised learning. The architecture of the second stage can emphasize understanding of what will happen in the current driving. After obtaining

the result from the second stage, additional information, such as crossing and hitting, as suspicious events can be obtained. Fusing the information from both metadata and event classification then gives rich information on the current driving situation. This approach enriches the event type by classifying the common event type on traffic risk incidents. The event type we can define is normal and anomaly, where the anomaly class has sub-classes for crossing and hitting. To achieve this, our technique is shown in Figure 4, where $\gamma_{trc}$ and $\gamma_{itc}$ are the model provided by [24].

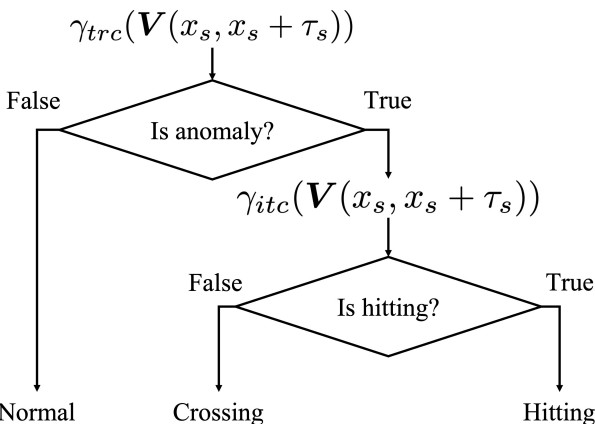

**Figure 4.** The proposed method architecture for the second stage is depicted in this diagram.

To create a robust model through the temporal dimension during training, our model uses the randomly temporal dimension length by the following.

$$\Xi = \{r : r = 1 + \lfloor \frac{random(\frac{\xi}{2}, 2\xi)}{\xi} \rfloor \times \omega \, , \; \omega \in \xi\}, \tag{8}$$

where $\xi$ is defined by 64 frames. As a result, the clipped video $V(x_s, x_s + \tau_s)$ has different fps after applying randomly temporal direction.

## 4. Experimental Results

In this experimental result, we describe the dataset used in this experiment. Then, the architecture of the proposed model is evaluated using different video classification methods. After evaluating those methods, the proposed model calculated the computational speed of our environment to run online.

### 4.1. Datasets

The experimental dataset used in this study is a contribution from [14] for training, validating, and testing datasets and a collection of real driving scenarios captured from various trucks over two months of testing. The explanation for each datapoint is shown by the following:

#### 4.1.1. Re-Annotated State-of-the-Art Traffic Risk Dataset

In this section, we will delve into the discussion regarding the information of the dataset used for training, validating, and testing, considering the classification method constructed by two distinct models: $\gamma_{trc}$ and $\gamma_{itc}$. To ensure consistency in dataset annotation, we employed the CST-S3D dataset [14]. The original video dataset comprises 704 clips, alongside two additional augmenting videos generated through conditional style translation. These augmenting videos converted scenes from day to night and vice versa. Consequently, the combined dataset contains 2112 videos, encompassing both the original footage and the augmented variations. For a more comprehensive understanding of the training models on $\gamma_{trc}$ and $\gamma_{itc}$, we provide a detailed explanation in the subsequent section.

CST-S3D Dataset to Build the Model of $\gamma_{trc}$

The construction of the $\gamma_{trc}$ model involves utilizing a dataset consisting of 1293 incident cases and 819 normal cases. This dataset is the foundation for training the model and evaluating its performance. The dataset is further divided into subsets for training, validation, and testing purposes to provide a more comprehensive breakdown. Specifically, for incident cases, 837 videos are allocated for training, 324 videos for validation, and 132 videos for testing. On the other hand, normal cases are represented by 612 training videos, 138 validation videos, and 69 testing videos. A detailed overview of the data distribution can be found in Table 1. The number of clipped videos is defined as the shifting of each frame, with the video length being $\xi$ frames as input of the second stage model.

**Table 1.** The detailed data information used in this experiment for training, validating, and testing the model of $\gamma_{trc}$ on both CST-S3D and real driving datasets. The definition of the number of clipped videos is that we shifted each video frame to 64 frames to be the input of our model.

| Dataset | | # of Videos | | # of Clipped Video (s) | |
|---|---|---|---|---|---|
| | | Incident | Normal | Incident | Normal |
| CST-S3D | Training | 837 | 612 | 26,097 | 33,060 |
| | Validating | 324 | 138 | 3264 | 3264 |
| | Testing | 132 | 69 | 7884 | 8724 |
| Real driving | Testing | 231 | 204 | 5001 | 5274 |

CST-S3D Dataset to Build the Model of $\gamma_{itc}$

The construction of the $\gamma_{itc}$ model relies on a dataset consisting of 528 incidents and 570 normal cases. This dataset is crucial for training and evaluating the performance of the model. Exploring the specifics, the dataset is divided into two main categories: incident and normal cases. Among the incident cases, there are 384 videos used for training, 66 videos for validation, and 132 videos for testing, specifically for hitting incident cases. On the other hand, for crossing incident cases, the dataset comprises 366 training videos, 54 validation videos, and 150 testing videos.

The distribution and organization of the dataset are further shown in Table 2, which provides a comprehensive breakdown of the data distribution. This table is a valuable resource, shedding light on the specific data used for training, validating, and testing purposes. Furthermore, it offers detailed insights into the objects involved in the incidents being tested, encompassing pedestrians, cyclists, motorbikes, cars, and trucks.

By incorporating this diverse dataset, comprising incidents and normal cases, along with the detailed data distribution provided in Table 2, the $\gamma_{itc}$ model can be built with a solid foundation. This extensive dataset facilitates the training, validation, and testing processes, enabling the model to learn and generalize from various scenarios and objects.

### 4.1.2. Real Driving

Specifically, the data are taken from the first and second months of the recording, with 17 and 5 days' worth of data, respectively. The dataset comprises 14,527 videos, with a total duration of 516 h. Each video varies between 2 and 3 min and has a dimension of $1280 \times 720$ pixels. The videos are saved in avi format, with a frame rate of 11 frames per second (fps).

An initial stage was run to identify potentially dangerous situations in the dataset, during which all possible hazardous scenarios were collected. Subsequently, the clipped videos were manually annotated, and the dataset was retested using the testing data with the number of incident annotation cases being 231 and 204 videos for the incident and normal cases shown in Table 1. To test the model of $\gamma_{itc}$, the real driving dataset creates 132 and 150 videos for hitting and crossing shown in Table 2.

**Table 2.** The detailed data information used in this experiment for training, validating, and testing the model of $\gamma_{itc}$ on both CST-S3D and real driving datasets. The definition of the number of clipped videos is that we shifted each video frame to 64 frames to be the input of our model.

| Dataset | Object Type | | # of Videos | | # of Clipped Video (s) | |
|---|---|---|---|---|---|---|
| | | | **Hitting** | **Crossing** | **Hitting** | **Crossing** |
| CST-S3D | Training | **All objects** | **384** | **366** | **11,052** | **9999** |
| | Validating | | **66** | **54** | **1602** | **1335** |
| | Testing | | **132** | **150** | **3987** | **2886** |
| | Detail for Testing | Pedestrian | 6 | 15 | 24 | 156 |
| | | Cyclist | 9 | 9 | 309 | 195 |
| | | Motorbike | 27 | 48 | 888 | 990 |
| | | Car | 15 | 6 | 729 | 174 |
| | | Truck | 75 | 72 | 2037 | 1371 |
| Real driving | Testing | **All objects** | **165** | **66** | **3885** | **1116** |
| | Detail for Testing | Pedestrian | 0 | 0 | 0 | 0 |
| | | Cyclist | 0 | 0 | 0 | 0 |
| | | Motorbike | 0 | 0 | 0 | 0 |
| | | Car | 54 | 39 | 1254 | 810 |
| | | Truck | 111 | 27 | 2631 | 306 |

*4.2. Result*

This section analyzes the results obtained from the proposed method for predicting traffic risk incidents. The analysis is conducted from the perspective of the temporal direction of time and also includes a comparison of the proposed method with different video classification methods. The result is separated into the output of the first and second stages for better analysis.

4.2.1. The First Stage Results

To begin the analysis, the results of the proposed method were examined in two different example driving scenarios as the result of the first stage. The near-miss incident cases obtained are shown in Figures 2 and 3. These figures visually represent the incident cases obtained by the proposed method, which can predict and highlight its ability to predict potential risks in real time.

A detailed analysis of the near-miss incident depicted in Figure 2 was conducted to better understand the proposed method's performance. The incident occurred between frames 271 and 349, as shown in the figure. The first stage result identified this incident as a near-miss because the car violated traffic rules by changing lanes from the left to the right side, which posed a potential danger to the driver on the road. From the driver's perspective, this event would have been hazardous and required quick reactions to avoid a collision. The driver in this scenario reacted by frequently braking to decrease the vehicle's speed and allow the car to cross safely into its lane.

Another incident analyzed in detail is depicted in Figure 3. This incident occurred between frames 172 and 195, and it was classified as a near-miss due to the potential danger it posed to the driver on the road. In this case, the car was stopped in the middle of the lane, creating a hazardous situation. The driver had to take evasive action to avoid crashing into the stationary car and could avoid the potential collision by changing lanes to the left. This situation highlights the importance of situational awareness and quick reactions on the road.

4.2.2. The Second Stage Result

Table 3 presents the performance metrics, including precision, recall, and F1-score, for the $\gamma_{trc}$ and $\gamma_{itc}$ methods on both the CST-S3D and real driving datasets. For the $\gamma_{trc}$

method, on the CST-S3D dataset, it achieved a precision of 90.99%, indicating that most of the predicted positive instances were true positives. The recall score was 86.98%, implying that the model successfully identified a high proportion of the actual positive instances in the dataset. The F1-score, which balances precision and recall, was calculated at 88.94%, representing a harmonic mean between the two metrics. This suggests that the $\gamma_{trc}$ method identifies true positives and minimizes false positives on the CST-S3D dataset.

**Table 3.** The performance of $\gamma_{trc}$ and $\gamma_{itc}$ methods on testing data for both CST-S3D and real driving datasets.

| Model | Dataset | Precision | Recall | F1-Score |
|---|---|---|---|---|
| $\gamma_{trc}$ | CST-S3D | 90.99 | 86.98 | 88.94 |
| | Real driving | 75.37 | 77.63 | 76.48 |
| $\gamma_{itc}$ | CST-S3D | 95.5 | 96.3 | 95.9 |
| | Real driving | 89.25 | 98.21 | 93.52 |

However, the $\gamma_{trc}$ method demonstrated slightly lower performance on the real driving dataset since the video quality and dashcam point of view differ from the CST-S3D dataset. It achieved a precision of 75.37%, implying a relatively higher number of false positive predictions. The recall score of 77.63% indicates that the model identified a significant portion of the actual positive instances. The F1-score of 76.48% suggests that the $\gamma_{trc}$ method's overall performance on the real driving dataset was moderate, balancing precision and recall.

Turning to the $\gamma_{itc}$ method, on the CST-S3D dataset, it exhibited a high precision of 95.5%, indicating a high number of correct positive predictions. The recall score was 96.3%, suggesting that the model successfully captured a large proportion of the true positive instances. The resulting F1-score of 95.9% demonstrates the robust performance of the $\gamma_{itc}$ method, striking a good balance between precision and recall on the CST-S3D dataset.

Similarly, the $\gamma_{itc}$ method demonstrated strong performance on the real driving dataset. It achieved a precision of 89.25%, indicating a relatively low number of false positive predictions. The recall score of 98.21% suggests that the model successfully identified almost all of the actual positive instances in the dataset. The resulting F1-score of 93.52% shows that the $\gamma_{itc}$ method's overall performance on the real driving dataset was high, reflecting a strong balance between precision and recall. The $\gamma_{trc}$ method performed well on the CST-S3D dataset, with high precision, recall, and F1-score. These results indicate that the $\gamma_{itc}$ method is more robust and reliable than the $\gamma_{trc}$ method, as it consistently performs well across different datasets.

Detailed performance for each classification class for 10 classes is shown in Table 4. The table presents the detailed performance of the mixed $\gamma_{trc}$ and $\gamma_{itc}$ video classification models on both the CST-S3D and real driving datasets. The performance metrics measured include precision, recall, and F1-score for different object types, categorized into hitting and crossing incident cases.

For the CST-S3D dataset, the precision, recall, and F1-score are reported for five object types: pedestrian, cyclist, motorbike, car, and truck. In the hitting incident case, the pedestrian class achieved a precision of 82.42%, recall of 78.79%, and F1-score of 80.56%. Similarly, for the crossing incident case, the pedestrian class obtained a precision of 86.68%, recall of 82.86%, and F1-score of 84.73%. The other object types, including cyclist, motorbike, car, and truck, also exhibited similar performance trends, with F1-scores ranging from 80.75% to 84.09% in the hitting incident case and from 81.27% to 82.95% in the crossing incident case. When considering the average performance across all object types, the mixed model achieved an average precision of 82.84%, recall of 79.19%, and F1-score of 80.97% in the hitting incident case. In the crossing incident case, the average precision was 84.68%, the recall was 80.95%, and F1-score was 82.77%.

**Table 4.** The detailed mixed performance of the $\gamma_{trc}$ and $\gamma_{itc}$ models on testing data for both CST-S3D and real driving datasets.

| Dataset | Object Type | Hitting | | | Crossing | | |
|---|---|---|---|---|---|---|---|
| | | Precision | Recall | F1-Score | Precision | Recall | F1-Score |
| CST-S3D | Pedestrian | 82.42 | 78.79 | 80.56 | 86.68 | 82.86 | 84.73 |
| | Cyclist | 83.19 | 79.52 | 81.31 | 83.14 | 79.48 | 81.27 |
| | Motorbike | 82.61 | 78.97 | 80.75 | 86.03 | 82.23 | 84.09 |
| | Car | 82.20 | 78.57 | 80.35 | 84.87 | 81.13 | 82.95 |
| | Truck | 83.79 | 80.10 | 81.90 | 82.7 | 79.05 | 80.83 |
| Average | | 82.84 | 79.19 | 80.97 | 84.68 | 80.95 | 82.77 |
| Real driving | Pedestrian | n/a | n/a | n/a | n/a | n/a | n/a |
| | Cyclist | n/a | n/a | n/a | n/a | n/a | n/a |
| | Motorbike | n/a | n/a | n/a | n/a | n/a | n/a |
| | Car | 66.93 | 67.17 | 67.05 | 74.37 | 76.98 | 75.65 |
| | Truck | 75.37 | 75.63 | 75.5 | 75.58 | 75.46 | 75.52 |
| Average | | 71.15 | 71.4 | 71.28 | 74.98 | 76.22 | 75.59 |

Moving on to the real driving dataset, the precision, recall, and F1-score were only reported for the car and truck object types since other objects had no incident. Consequently, the metrics were unavailable for the pedestrian, cyclist, and motorbike classes. In the hitting incident case, the car class achieved a precision of 66.93%, recall of 67.17%, and F1-score of 67.05%. Similarly, in the hitting incident case, the car class obtained a precision of 74.37%, recall of 76.98%, and F1-score of 75.65%. The truck class exhibited slightly better performance, with an F1-score of 75.5% and 75.52% in the hitting and crossing incident cases, respectively. When considering the average performance for the available object types, the mixed model achieved an average precision of 71.15%, recall of 71.4%, and F1-score of 71.28% in the hitting incident case. In the hitting incident case, the average precision was 74.98%, the recall was 76.22%, and F1-score was 75.59%. From the above analysis, the mixed $\gamma_{trc}$ and $\gamma_{itc}$ video classification models demonstrated promising performance on both the CST-S3D and real driving datasets. They achieved high precision, recall, and F1-score across different object types and scenarios. These results indicate the potential of the mixed model in accurately classifying objects and distinguishing between hitting and crossing incidents in various driving scenarios.

*4.3. Computational Time*

To evaluate the efficiency of the proposed methodology, a computational time analysis was performed by randomly selecting a video containing 2110 frames. This analysis was conducted to measure the performance of the proposed method in terms of computational complexity and execution time. It showed that the proposed method has an average frame rate of 9.60, with a standard deviation of 1.44. In addition to the average frame rate, the analysis also yielded information on the maximum and minimum fps that the proposed method can achieve. The maximum fps reached by the method was 14.28, while the minimum fps was 7.35. These values indicate the range of performance expected from the proposed method under different circumstances. Ensuring the results of the computational time analysis were reliable and accurate, detailed information was provided regarding the hardware and software environment used to run the proposed method. This information was presented in Table 5 and included details about the type and specifications of the computer used and the software programs and libraries utilized in the analysis. Based on the current computational speed of the proposed algorithm, it is possible to use the online version of the method. The proposed method can be implemented in real-time applications such as surveillance system predictions on the dash camera. The ability to run the method online indicates that it is efficient and can process data quickly, which is essential in applications where time is critical.

**Table 5.** Hardware and software environment for running proposed method.

| | Spesifications | |
|---|---|---|
| | CPU | 11th Gen Intel(R) Core(TM) i9-11900K @ 3.50 GHz |
| Hardware | RAM | 64 GB |
| | GPU | NVIDIA GeForce RTX 3090 |
| | OS | Windows 11 Pro 22H2 |
| Software | IDE | Microsoft Visual Studio Enterprise 2019 |
| | Language | Python 3.9.7 |
| | DL tools | Torch 1.10.1 + CUDA 11.3 |

## 5. Theoretical and Managerial Implication

This research proposed a faster and more accurate algorithm for predicting traffic risk incidents and monitoring near-miss events with limited data annotation, where our system archives the evaluation performance on both CST-S3D and real driving datasets. The study utilized unsupervised and supervised learning techniques to identify high-risk incidents in online videos. This approach aims to improve the accuracy and efficiency of identifying potential risks and help prevent future incidents. Furthermore, the proposed method has the potential to assist distribution companies in evaluating the driving performance of their employees. By monitoring near-miss events and analyzing driving patterns, companies can identify areas for improvement in their driver training programs and reduce the likelihood of future accidents. In other words, the proposed method enhances the classification process with data limitations in the first-person dashcam video. Since our method successfully classified the traffic risk incident prediction, it can be useful to give feedback or evaluation the drivers' performance and how safe they drive. By automatically predicting the risky event, the company does not need to monitor its drivers by manually checking how well they drive.

## 6. Conclusions and Discussion

In this paper, we proposed two-stage approaches based on unsupervised and supervised learning for predicting the incidental scenario running on the online version. Our approach relied on the human perception of driving to anticipate and identify risky events on the road. However, one primary challenge is creating an algorithm to produce an end-to-end online version of traffic risk prediction. To address this challenge, we incorporated two critical pieces of information into our prediction algorithm: the safety and real distances that should be maintained between the driver and other moving objects. By leveraging these two pieces of information, we could accurately predict when a potentially risky event might occur and localize the object most likely to cause a safety violation. To further improve the accuracy and reliability of our predictions, we also utilized the classification of incident cases on different types of incidents using supervised learning. This learning involved using models explicitly designed to classify the output generated by our prediction algorithm from the first stage to enrich the types of incidents. By applying these techniques, we achieved high accuracy and provided a more reliable tool for predicting and mitigating traffic risk with various incident classification types.

The proposed method was able to predict and classify the near-miss incident accurately. It addresses several challenges to mitigating the traffic risk incident by providing early warning and classifying it into common incident types and risky objects. Providing early warning gives significant advantages to giving the driver evaluation and recommendation feedback based on event collections. Analyzing the event in detail makes it possible to gain a deeper understanding of the proposed method's performance and its ability to predict potential risks on the road. These types of incidents can have severe consequences if there is no action from the driver or other moving objects, including injury or loss of life, making it crucial to develop effective methods for predicting and preventing accidents on the road. The proposed method has the potential to play a significant role in improving road safety

by identifying potential risks in real time and alerting drivers to take appropriate action. This incident analysis provides valuable insight into the proposed method's performance and ability to accurately predict potential road risks based on safety distance.

The effectiveness of traffic risk classification can be attributed to successfully utilizing the supervised learning model by transferring metadata to enrich the classification type. This learning allows for more efficient and accurate incident classification instead of relying solely on collecting huge amounts of video annotations, which can be time-consuming and prone to error. Evaluating these methods highlights the importance of utilizing advanced machine-learning techniques for traffic incident classification. More efficient and accurate incident prediction can be achieved by leveraging these techniques, improving road safety.

### 7. Limitation and Future Works

Even though the proposed method's computational time can run through an online version, its use has certain limitations, which are currently restricted to dashcam attachments on trucks with high vision capabilities. However, despite these limitations, the proposed method has shown promising results in terms of its ability to process data from dashcams in real time, making it a valuable tool for enhancing the safety and efficiency of truck operations.

To further expand the capabilities of the proposed method, future research will focus on creating an algorithm that can be adapted for use on different types of ego-vehicles, such as passenger cars, buses, and even autonomous vehicles. This will involve extensive testing and optimization to ensure the algorithm can accurately process and analyze data from various video perspectives for each ego-vehicle. Additionally, the research will explore the possibility of integrating the proposed method with other existing technologies, such as 5G networks, Internet of Things (IoT) devices, and vehicle-to-vehicle communication systems, to enhance the overall performance and effectiveness of the method. This could enable the algorithm to leverage additional data sources and improve its decision-making capabilities, improving safety, efficiency, and traffic management results. These integrating systems could enable the algorithm to operate in real time with reduced latency, enabling faster and more accurate detection and prediction of potential hazards and obstacles on the road.

Furthermore, the research will also investigate the potential of leveraging machine learning techniques further to enhance the accuracy and reliability of the proposed method. By collecting large datasets of real-world driving data and training the algorithm on them, we aim to improve its ability to accurately detect and predict potential hazards and obstacles on the road and optimize its decision-making capabilities in complex driving scenarios. Another important aspect of the future research direction will be integrating the proposed method into existing transportation infrastructure and regulations. This will involve working closely with industry stakeholders, policymakers, and other relevant parties to ensure the algorithm complies with safety standards, privacy regulations, and ethical considerations. This may also involve developing guidelines and best practices for using the algorithm in different settings, such as urban environments, rural areas, and highways.

Moreover, the research will explore the potential of the proposed method for use in other related domains, such as smart city applications, intelligent transportation systems, and fleet management. This could involve adapting the algorithm for use in different contexts, such as public transportation, delivery trucks, and emergency vehicles, to further enhance the safety and efficiency of these operations. The research will also focus on evaluating the economic and societal impacts of the proposed method. This will involve conducting cost-benefit analyses, evaluating the potential savings in reduced accidents, improved traffic flow, and enhanced fuel efficiency, and assessing the potential social and environmental benefits of safer and more efficient road transportation. These evaluations will help inform decision-makers and stakeholders about the potential value and viability of implementing the proposed method on a larger scale.

**Funding:** This research received no external funding.

**Institutional Review Board Statement:** Not applicable.

**Informed Consent Statement:** Not applicable.

**Data Availability Statement:** Not applicable.

**Conflicts of Interest:** The author declares no conflict of interest.

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
