# Peer review of "An End-to-End Online Traffic-Risk Incident Prediction in First-Person Dash Camera Videos"

_2504-2289, doi:10.3390/bdcc7030129_

Round 1

Reviewer 1 Report

This paper presents a framework to predict any incident that could cause road safety issues. The author proposes a two-stage architecture model: in the first stage, unsupervised learning is applied to collect all the suspicious events, where in the second stage a supervised learning model applied to classify those events. The paper is very well structured; with a detailed introduction, where the contribution of this study is clearly presented, followed by a thorough literature review. The presented methodology has scientific validity and successfully conveyed to the reader. The author provides the complete framework on the subject with adequate volume of data, method, and well founded and interesting results. Some minor issues should be tackled before the paper is accepted for publication:

In my opinion, it would be useful to slightly expand the "Methods" section, and in particular the subsection 3.3, by providing more detailed information on the model developed.

Firstly, few typos as well as grammar and syntactical errors should be doubled checked to make the text flow more naturally. For instance, the section in lines 5 & 6 (concerning the research questions), can be improved in order to make it more comprehensible to the reader.

Lastly, all occurrences of the term “accident” in the whole manuscript should be replaced with the term “crash”. The authors are urged to choose 'crash' over 'accident' and use the term uniformly throughout the paper to avoid confusion. 'Accident' implies an unavoidable act that cannot be helped, while 'crash' simply describes the collision act.

Reviewer 2 Report

The manuscript follows very well the main sections of a properly written research article, which are also reflected concisely in the abstract. It is easy to read and to follow what the authors proposed to tackle, setting up (in the introduction section) two hypotheses, which they later tested out and quantified their success rate. The architecture of the proposed system has a nicely described mathematical model, which seems to be correct and it’s checkable. The authors conducted a literature review, that not only places very well the manuscript in a certain field (area of interest), by exploring current limitations of existing similar solutions, but also refer to relevant works in that particular field and they are sufficient in quantity. The results are presented in an easy-to-follow manner and the authors even go so far as to describe a few limitations of their approach (and future directions). The conclusions seem to be supported by the results the authors obtained. Finally, I also appreciate the quality of the English language and grammar of the manuscript (almost flawless).
I just have one simple remark for the authors: the captions of the tables and figures should not include any extensive descriptions. Those should be in separate paragraphs (before or after the figures and tables) and the captions should be limited to one, or two lines (at the most).
I congratulate the authors for their work!
The manuscript follows very well the main sections of a properly written research article, which are also reflected concisely in the abstract. It is easy to read and to follow what the authors proposed to tackle, setting up (in the introduction section) two hypotheses, which they later tested out and quantified their success rate. The architecture of the proposed system has a nicely described mathematical model, which seems to be correct and it’s checkable. The authors conducted a literature review, that not only places very well the manuscript in a certain field (area of interest), by exploring current limitations of existing similar solutions, but also refer to relevant works in that particular field and they are sufficient in quantity. The results are presented in an easy-to-follow manner and the authors even go so far as to describe a few limitations of their approach (and future directions). The conclusions seem to be supported by the results the authors obtained. Finally, I also appreciate the quality of the English language and grammar of the manuscript (almost flawless).
I just have one simple remark for the authors: the captions of the tables and figures should not include any extensive descriptions. Those should be in separate paragraphs (before or after the figures and tables) and the captions should be limited to one, or two lines (at the most).
I congratulate the authors for their work!The manuscript follows very well the main sections of a properly written research article, which are also reflected concisely in the abstract. It is easy to read and to follow what the authors proposed to tackle, setting up (in the introduction section) two hypotheses, which they later tested out and quantified their success rate. The architecture of the proposed system has a nicely described mathematical model, which seems to be correct and it’s checkable. The authors conducted a literature review, that not only places very well the manuscript in a certain field (area of interest), by exploring current limitations of existing similar solutions, but also refer to relevant works in that particular field and they are sufficient in quantity. The results are presented in an easy-to-follow manner and the authors even go so far as to describe a few limitations of their approach (and future directions). The conclusions seem to be supported by the results the authors obtained. Finally, I also appreciate the quality of the English language and grammar of the manuscript (almost flawless).
I just have one simple remark for the authors: the captions of the tables and figures should not include any extensive descriptions. Those should be in separate paragraphs (before or after the figures and tables) and the captions should be limited to one, or two lines (at the most).
I congratulate the authors for their work!
The manuscript follows very well the main sections of a properly written research article, which are also reflected concisely in the abstract. It is easy to read and to follow what the authors proposed to tackle, setting up (in the introduction section) two hypotheses, which they later tested out and quantified their success rate. The architecture of the proposed system has a nicely described mathematical model, which seems to be correct and it’s checkable. The authors conducted a literature review, that not only places very well the manuscript in a certain field (area of interest), by exploring current limitations of existing similar solutions, but also refer to relevant works in that particular field and they are sufficient in quantity. The results are presented in an easy-to-follow manner and the authors even go so far as to describe a few limitations of their approach (and future directions). The conclusions seem to be supported by the results the authors obtained. Finally, I also appreciate the quality of the English language and grammar of the manuscript (almost flawless).
I just have one simple remark for the authors: the captions of the tables and figures should not include any extensive descriptions. Those should be in separate paragraphs (before or after the figures and tables) and the captions should be limited to one, or two lines (at the most).
I congratulate the authors for their work!

Reviewer 3 Report

An End-to-End Online Traffic-Risk Incident Prediction in First-Person Dash Camera Videos” is the title of the author’s work. Authors should respond to the following questions and incorporate their answers into their manuscript as remarks with additional details.

1.     What is the significance of predicting traffic risk incidents in first-person dash camera videos?

2.     How does the proposed two-stage architecture address the challenge of early warning and accurate perception in traffic-risk prediction?

3.     What are the advantages of using unsupervised learning in the first stage of the architecture?

4.     How does the second stage of the architecture leverage supervised learning to classify suspicious events?

5.     How does the inclusion of metadata from the first stage enhance the classification process in the second stage?

6.     What is the average frame rate achieved by the proposed method during online situations, and why is this important?

7.     How do the F1-scores on labeled data of CST-S3D and real driving datasets demonstrate the effectiveness of the proposed method?

8.     How can the proposed method benefit distribution companies in evaluating the driving performance of their drivers?

9.     In what ways can the automatic monitoring of near-miss events and analysis of driving patterns contribute to reducing future accidents?

10.  What are the potential future applications of the research presented in this paper beyond traffic-risk incident prediction?

No comments

Round 2

Reviewer 3 Report

No further comments

No comments